# NetMoE: Accelerating MoE Training through Dynamic Sample Placement

**Xinyi Liu**[1] **Yujie Wang**[1] **Fangcheng Fu**[1] **Xupeng Miao**[2] **Shenhan Zhu**[1]
**Xiaonan Nie**[1] **Bin Cui**[1,3]
[1]School of CS & Key Lab of High Confidence Software Technologies (MOE), Peking University
[2]Purdue University   [3]Institute of Computational Social Science, Peking University (Qingdao)
`{xy.liu, alfredwang, ccchengff}@pku.edu.cn`
`xupeng@purdue.edu`,`{shenhan.zhu, xiaonan.nie, bin.cui}@pku.edu.cn`

## Abstract

Mixture of Experts (MoE) is a widely used technique to expand model sizes for better model quality while maintaining the computation cost constant. In a nutshell, an MoE model consists of multiple experts in each model layer and routes the training tokens to only a fixed number of experts rather than all. In distributed training, as experts are distributed among different GPUs, All-to-All communication is necessary to exchange the training tokens among the GPUs after each time of expert routing. Due to the frequent and voluminous data exchanges, All-to-All communication has become a notable challenge to training efficiency.

In this paper, we manage to accelerate All-to-All communication in MoE models from the training sample perspective, which is unexplored so far. In particular, we put forward the observation that tokens in the same training sample have certain levels of locality in expert routing. Motivated by this, we develop NetMoE, which takes such locality into account and dynamically rearranges the placement of training samples to minimize All-to-All communication costs. Specifically, we model the All-to-All communication given the sample placement and formulate an integer programming problem to deduce the optimal placement in polynomial time. Experiments with 32 GPUs show that NetMoE achieves a maximum efficiency improvement of $1.67\times$ compared with current MoE training frameworks.

## 1 Introduction

In recent years, large language models (LLMs) have shown impressive performance in language understanding and generation (OpenAI, 2023; Touvron et al., 2023; Zhou et al., 2024; Dubey et al., 2024; Shao et al., 2024; Zhang et al., 2024a) due to the increasing model size. However, larger models often come with greater computational costs. To address this, Mixture of Experts (MoE) models have been introduced to expand the model size greatly without increasing the computational cost. Combining MoE with Transformer-based models can yield outstanding performance across various tasks, including natural language processing (Lepikhin et al., 2021; Fedus et al., 2022),

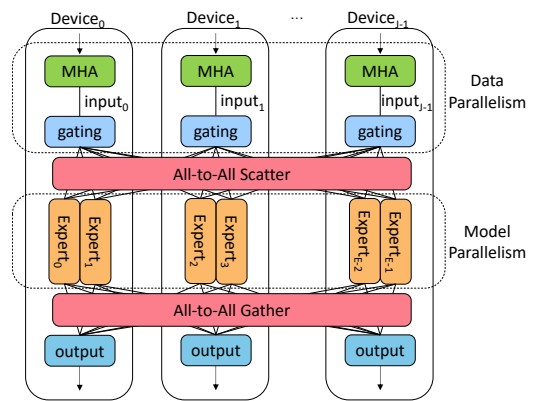

Figure 1: An example of expert parallelism applied to an MoE model with $J$ devices and $E = 2J$ experts (each device has two different experts).

computer vision (Riquelme et al., 2021; Liang et al., 2022), recommendation systems (Tang et al., 2020; Zou et al., 2022), and speech recognition (You et al., 2022; Kwon & Chung, 2023).

MoE models often replace the feed-forward network (FFN) layer with the MoE layer, which consists of a gating network and several small FFNs, representing different experts. In the MoE layer, each token is routed by the gating network to only a few selected experts, and the final output is

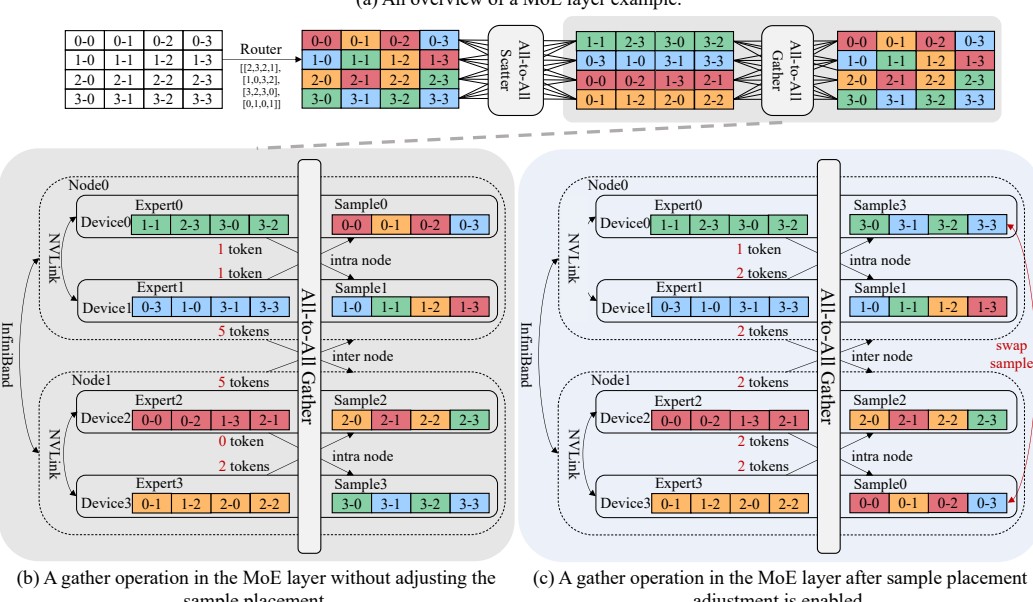

Figure 2: An example of sample exchange. The figure illustrates the All-to-All gather operation in a MoE layer with two nodes, each containing two devices, and each device having one expert. Different colors represent tokens sent to different experts, and $i$-$j$ denotes the $j$-th token in the $i$-th sample. Fig. 2(a) illustrates the complete process of a MoE layer during forward propagation. Fig. 2(b) shows the All-to-All gather operation in the MoE layer without adjusting the sample placement, where the inter-node communication volume of each node is 5 tokens. Fig. 2(c) displays the All-to-All gather operation after sample placement adjustment is enabled — the positions of samples on the devices change (samples 0 and 3 are exchanged), reducing the inter-node communication volume to 2 tokens per node.

obtained by a weighted sum of the computations from the selected experts. By such means, we can increase the number of experts to expand the model size for better performance, while keeping the computation complexity constant.

Despite the above benefit, given the potentially large number of experts, the memory capacity of a single device is often insufficient. As a result, expert parallelism (Lepikhin et al., 2021; Fedus et al., 2022) is a common technique to facilitate the distributed training of MoE models. As shown in Fig. 1, each device holds only a subset of the experts to reduce memory consumption. Meanwhile, other model parameters are replicated and stored on all devices, and the training data assigned to each device are different. In each MoE layer, based on the routing result of the gating network, each token is sent to the device where the selected expert is located. The output from the expert is then sent back to the original device of the corresponding token. This involves two communication operations, namely the All-to-All scatter and All-to-All gather (He et al., 2021), respectively.

Due to the dynamic nature of routing, training MoE models efficiently faces several challenges, with the All-to-All communication, being the most significant one. Particularly, the All-to-All communication time can account for up to 80% of the total training time (Hwang et al., 2023; Liu et al., 2023; He et al., 2022; Li et al., 2023; Yu et al., 2024). One reason is because all tokens need to participate in the All-to-All operation, leading to a high communication volume. Another reason is the communication frequency. Considering both the forward and backward propagation, each MoE layer requires four All-to-All communications per training iteration. Such frequent and extensive communication incurs significant time costs. Therefore, accelerating All-to-All communication is essential to improve training efficiency.

**Motivation:** Recent studies have demonstrated that expert routing exhibits a certain degree of *data locality*. To be specific, input tokens may have distinct preferences for experts, and the corresponding distribution is often skewed (He et al., 2022; Nie et al., 2023; Xue et al., 2024; Jiang et al., 2024). In other words, the All-to-All operation in MoE models can be highly unbalanced across different devices, thus bounded by the device pair with the highest communication volume. Mean-

while, it is well known that *network locality* is an inherent characteristic of modern clusters for deep learning training. In particular, there are various communication channels in modern clusters, e.g. intra-node devices usually communicate via PCIe or NVLINK, while inter-node devices use Ethernet or InfiniBand, with intra-node communication usually faster than inter-node ones. To achieve load balancing, existing methods propose techniques from the model perspective (Nie et al., 2023; Lewis et al., 2021). Typically, they either dynamically adjust the model placement but introduce a lot of additional communication, or modify the model definition but sacrifice the model performance (see §2.2 for more discussion). Yet optimization from the data perspective is under explored. Inspired by this, we propose NetMoE, which accelerates the All-to-All communication by combining the data locality in expert routing with the network locality among training devices. The essential idea of NetMoE is to dynamically adjust the placement of data samples during training based on expert routing results so that more tokens will be transmitted through high-speed channels rather than low-speed ones. As illustrated in Fig. 2, $sample_0$ shows a preference for $expert_2$ that resides on $node_1$, while $sample_3$ favors $expert_0$ that resides on $node_0$. Using the vanilla All-to-All communication method would result in significant inter-node communication overhead, as shown in Fig. 2(b). However, by swapping the positions of $sample_0$ and $sample_3$ as depicted in Fig. 2(c), part of the inter-node communication can be converted into intra-node communication or even intra-device memory copying, significantly reducing the time cost (detailed in §3.1). In this way, we can accelerate the All-to-All communication without affecting the computing results.

However, it is non-trivial to achieve dynamic sample placement. For one thing, how to adjust the placement to maximize efficiency is a complex and unexplored question. For another, since the adjustment should be done for every layer in every iteration, it is vital to devise an efficient algorithm to deduce the placement on the fly. To address these problems, we first revisit the cost modeling for All-to-All communication and formulate the dynamic sample placement problem into a combinatorial optimization problem. Subsequently, we split it into two stages to ease the solving and design a corresponding polynomial-time algorithm to ensure a timely solution.

In short, the technical contributions of this work are summarized as follows:

- We propose NetMoE, the first effort that leverages both the data locality and network locality to accelerate the All-to-All communication through dynamic sample placement.

- We formulate the dynamic sample placement problem as a combinatorial optimization problem, which aims to find the best sample placement that maximizes efficiency given the expert routing.

- We dissect the problem into two stages and develop a polynomial-time solution to efficiently derive the sample placement during training.

- We conduct experiments with various models on 32 NVIDIA A800 GPUs. Results show that NetMoE outperforms current MoE training systems by up to $1.67\times$ in terms of training efficiency.

## 2 PRELIMINARY

### 2.1 PARALLELISM IN DISTRIBUTED TRAINING

**Data and Model Parallelism:** In data parallelism (Li et al., 2020; Sergeev & Balso, 2018; Wang et al., 2023; Zhang et al., 2024b), each device maintains a complete copy of the model parameters, while different training samples are assigned to each device. After the backward computation is completed, the model gradients from all devices are aggregated before updating the model parameters. In model parallelism (Narayanan et al., 2021b; Huang et al., 2019; Narayanan et al., 2021a; Guan et al., 2024), model parameters are distributed across multiple devices, with each device responsible for only a portion of the model. Communication operations are necessary to transmit the intermediate results (a.k.a. forward activations and their backward gradients) to accomplish the forward and backward propagation.

**Expert Parallelism:** As shown in Fig. 1, expert parallelism (Lepikhin et al., 2021; Fedus et al., 2022) can be regarded as combining model parallelism and data parallelism. It distributes expert parameters across different devices like model parallelism, while replicating other parameters on all devices like data parallelism. In each MoE layer, each token will be routed by the gating network to top $K$ different experts for processing, where $K$ is a hyperparameter, typically a small value, such as 1 or 2, which helps to reduce the computational complexity. After the MoE layer obtains the

gating routes, tokens are sent to the devices where the corresponding experts are located based on the routing. The results from the expert computations are then sent back to the original devices where the tokens are located. Since the experts are distributed across different devices, communication during this process involves all devices sending and receiving messages with one another, leading to what is known as All-to-All communication.

## 2.2 DISTRIBUTED TRAINING ACCELERATION TECHNIQUES FOR MoE MODELS

**Dynamic Expert Placement:** The efficiency of MoE models is constrained by the extensive and frequent All-to-All communication required during training. In response to this issue, some studies have observed that data tends to show a preference for certain experts during training. Then, based on this observation, they further propose to dynamically adjust the placement of experts to reduce communication volume (He et al., 2022; Nie et al., 2023; Zhai et al., 2023). For instance, popular experts can be placed on more devices in the data parallel manner, so that the communication volume related to them would decrease. However, due to the growing size of experts, these approaches incur substantial overhead of transmitting expert parameters among the devices, so they cannot adjust the expert placement for every iteration, leading to sub-optimality. In contrast, our work tries to reduce the communication volume from a different perspective: we dynamically adjust the placement of samples in every iteration to accelerate the All-to-All communication. To be specific, we formulate an optimization problem to deduce the best sample placement that minimizes the time cost of All-to-All communication. As we will evaluate in §4, our work outperforms existing works based on dynamic expert placement when training MoE models.

**Modification in Model Definition:** To achieve better workload balance in MoE training, there are many existing works developed to modify the model definition (e.g., routing mechanisms, model architectures). Some approaches modify the routing mechanism to balance the load across experts, which helps reduce synchronization time between devices (Lewis et al., 2021). Recognizing the network locality in distributed training, several works introduce a routing topology loss to prioritize routing tokens within the same node, thereby reducing inter-node communication (Li et al., 2024; Chen et al., 2022). Other approaches (Zeng & Xiong, 2023) map tokens to smaller hidden layer dimension before inter-node communication, further decreasing the communication load. SC-MoE (Cai et al., 2024) proposes feeding the output of the current attention layer directly into the next MoE layer, enabling parallel forward propagation with the current MLP layer in order to fully overlap All-to-All communication with computation. Although these methods improve training efficiency, they inevitably impact model convergence.

When applying these methods, we usually need to run numerous trials to tune the hyper-parameters, such as adjusting the weight of the topology-aware routing loss (Chen et al., 2022) or tuning the hyper-parameters for different communication channels (Zeng & Xiong, 2023). Given that each trial of LLM training can take days or even months, their utility is inevitably hampered. In contrast, our work focuses on how to accelerate All-to-All communication without affecting model convergence.

Figure 3: The overview of the method of NetMoE.

## 3 NETMOE

In this section, we introduce NetMoE, a novel framework designed to optimize distributed training for MoE models by considering both data and network locality. Given a target MoE model and the hardware environment, NetMoE aims to minimize the All-to-All communication cost. Its core innovation lies in optimizing the placement of samples within each MoE layer to maximize the utilization of faster intra-node bandwidth, thereby reducing the communication volume over slower inter-node connections. Specifically, NetMoE swaps the samples across devices during each MoE layer, enabling more tokens to communicate within the node during All-to-All communication.

Table 1: Notations used throughout this work. We assume $I$ is divisible by $J$, and $J$ is divisible by $N$, which are common in distributed training.

| | |
|---|---|
| $L$ | The number of tokens per sample. |
| $H$ | The hidden size for each token. |
| $E$ | The number of experts in the MoE layer. |
| $K$ | The number of experts to be routed per token. |
| $I$ | The number of samples per iteration (a.k.a. global batch size). |
| $J$ | The number of devices (i.e., GPUs). |
| $N$ | The number of nodes (machines). |
| $\mathbb{I}[\cdot]$ | The indicator function. |
| $[\![n]\!]$ | The set of natural numbers less than $n$, i.e., $\{0, 1, \cdots, n-1\}$. |

Table 2: Bandwidth of each channel of the NVIDIA A800 GPU cluster used in our experiments.

| Channel | Bandwidth |
|---|---|
| Intra-device | $\sim$2TB/s |
| Intra-node | 400GB/s |
| Inter-node | 100GB/s |

Fig. 3 illustrates the overview of this section. We begin by introducing the modeling of All-to-All communication in MoE training and formulate our optimization problem in §3.1. We then illustrate how to solve the problem in §3.2, with the detailed algorithm shown in Alg. 1. We also present our implementation details in §3.3. For clarity, the frequently used notations are listed in Table 1.

### 3.1 PROBLEM FORMULATION

**Communication Modeling:** We first discuss the mathematical modeling of All-to-All communication, which is the optimization target of NetMoE. We use the $\alpha$-$\beta$ model (Sarvotham et al., 2001) to analyze All-to-All communication, where $\alpha$ represents the latency cost and $\beta$ represents the bandwidth cost. Specifically, we classify communication into three categories: intra-device, intra-node, and inter-node communication, each using different channels. Table 2 lists the bandwidth of each channel used in our experiments. Since intra-device communication is typically achieved via memory copying, it is significantly faster than the other two categories and thus not considered in our modeling. Therefore, the communication time is determined by the maximum time required for data transfer across the intra-node and inter-node channels. The bandwidths of these channels are represented by $v_{intra}$, and $v_{inter}$, respectively. Thus, for each All-to-All communication, its time cost can be expressed by the following formula, where $s.$ represents the communication volume for the corresponding channel.

$$t = \max(t_{intra}, t_{inter}), \quad \text{where} \quad \begin{aligned} &t_{intra} = \alpha_{intra} + \beta_{intra}s_{intra}, \beta_{intra} = 1/v_{intra}, \\ &t_{inter} = \alpha_{inter} + \beta_{inter}s_{inter}, \beta_{inter} = 1/v_{inter} \end{aligned} \tag{1}$$

The bandwidth ($v.$) and latency ($\alpha.$) can be obtained by profiling the hardware environment before training, while the communication volume ($s.$) needs to be dynamically determined based on the routing results within the MoE layer. We then analyze how to calculate the communication volume.

Let $route \in \mathbb{N}^{I \times L \times K}$ be the token routing results of the gating network, which represents the $K$ experts that each token will be sent to. Then, the number of tokens that the $i$-th sample needs to send to the $e$-th expert can be counted as

$$num_{i,e} = \sum_{l,k} \mathbb{I}[route_{i,l,k} = e] \text{ for } i \in [\![I]\!], e \in [\![E]\!] \tag{2}$$

Next, $num \in \mathbb{N}^{I \times E}$ can be used to model the communication volume across different channels. Let $\texttt{ExpDev}(e)$ be the device index of the $e$-th expert, $\texttt{SmpDev}(i)$ the device index where the $i$-th sample should be routed to, and $\texttt{Node}(j)$ the node index of the $j$-th device. By considering the communication volume as the number of tokens that need to be transmitted, we have

$$s_{intra} = \sum_{(i,e) \in S_{intra}} num_{i,e}, \quad s_{inter} = \sum_{(i,e) \in S_{inter}} num_{i,e} \tag{3}$$

where $S_{intra}$ and $S_{inter}$ can be calculated via the device indices of experts and samples:

$$\begin{aligned} S_{intra} &= \{(i,e)|\texttt{Node}(\texttt{SmpDev}(i)) = \texttt{Node}(\texttt{ExpDev}(e)) \wedge \texttt{SmpDev}(i) \neq \texttt{ExpDev}(e)\} \\ S_{inter} &= \{(i,e)|\texttt{Node}(\texttt{SmpDev}(i)) \neq \texttt{Node}(\texttt{ExpDev}(e))\} \end{aligned} \tag{4}$$

**Rationality of Dynamic Sample Placement:** Given the aforementioned modeling, there is no doubt that the time cost of All-to-All communication is highly related to the placement of experts and

samples. In practice, dynamically adjusting the placement does not affect the training results as the All-to-All communication is still correctly performed. Combining with the common fact of network locality that $v_{intra} > v_{inter}$, we can adjust the placement of samples and/or experts to reduce the inter-node communication volume, even if the intra-node communication volume becomes slightly higher. In fact, with a similar goal, several existing works have proposed to dynamically adjust the placement of experts based on their popularities (He et al., 2022; Nie et al., 2023; Zhai et al., 2023), as introduced in §2. However, all these works overlook the data locality — tokens of the same sample are usually routed to the same expert (Xue et al., 2024; Jiang et al., 2024), thereby missing the optimization opportunity of dynamic sample placement. More importantly, the size of the parameters of experts is usually much larger than the size of the samples. This prevents previous works from adjusting the expert placement in every iteration. In contrast, the adjustment of sample placement can be fused with the All-to-All communication by nature (detailed below), requiring zero extra communication. Consequently, this work focuses on the unexplored aspect, aiming to accelerate MoE training by dynamically adjusting the sample placement.[1]

To help readers better understand the strength of dynamic sample placement, we take Fig. 2 as an example, where $I = 4, L = 4, E = 4, K = 1$, and both the experts and samples are placed sequentially, i.e., $\texttt{ExpDev} = [0, 1, 2, 3]$, $\texttt{SmpDev} = [0, 1, 2, 3]$. Fig. 2(b) shows the communication without changing the placement of samples. According to Eq. 3 and Eq. 4, if we only consider the sending volume of node 0, then $S_{inter} = \{(0, 2), (0, 3), (1, 2), (1, 3)\}$, indicating that $s_{inter} = 5$. However, after optimizing the placement of samples as in Fig. 2(c), i.e., $\texttt{SmpDev} = [3, 1, 2, 0]$, the corresponding inter-node communication volume changes into $S_{inter} = \{(3, 2), (3, 3), (1, 2), (1, 3)\}$, which gives $s_{inter} = 2$. Furthermore, it is worth noting that the sample placement adjustment can be combined with the All-to-All gather operation. To be specific, instead of restoring tokens to their original positions, they are directly placed in their new positions according to the altered sample placement. This method directly optimizes the current communication operation without introducing any extra communication.

**Problem Formulation:** After the sample placement adjustment is determined, it can be seen that altering $\texttt{SmpDev}$ affects two All-to-All operations: the gather operation of the current MoE layer and the scatter operation of the next MoE layer. Thus, our optimization targets these two operations. For the $l$-th layer, the optimization problem can be written as follows.

$$
\underset{\texttt{SmpDev}(i) \in [\![J]\!] \text{ for } i \in [\![I]\!]}{\arg\min} \quad t^{(l,gather)} + t^{(l+1,scatter)}
$$
$$
= \max\left(t_{intra}^{(l,gather)}, t_{inter}^{(l,gather)}\right) + \max\left(t_{intra}^{(l+1,scatter)}, t_{inter}^{(l+1,scatter)}\right) \quad (5)
$$
$$
\text{s.t.} \quad \sum_{i \in [\![I]\!]} \mathbb{I}[\texttt{SmpDev}(i) = j] = I/J \text{ for } j \in [\![J]\!]
$$

Since a single change in sample placement affects two All-to-All operations, both communication times are included in the optimization objective. Additionally, to ensure computational and memory balance across devices, each device should retain the same number of samples before and after the sample placement adjustment. This forms the basis for the constraints in our optimization.

## 3.2 PROBLEM SOLVING

Eq. 5 is a complex combinatorial optimization problem, which cannot be solved optimally in polynomial time. As the cluster size increases, even finding an approximate solution may take a significant amount of time. Since this problem needs to be solved before each gather operation, solving it directly would result in unbearable additional time costs. To address this, we design an efficient method to obtain approximate solutions. In particular, we first dissect the optimization problem into two stages and develop a polynomial-time algorithm to achieve the solution, as introduced below.

**Two-Stage Dissection:** Although Eq. 1 takes the maximum value of the two kinds of communication cost, in practice, due to the significant bandwidth difference between the inter- and intra-node

---

[1]Our work is fully compatible with the dynamic expert placement technique. Specifically, in the problem formulation and solving of NetMoE, we do not make any assumption on the expert placement. Instead, it is treated as an input. Thus, we can dynamically adjust the expert placement like previous works, and NetMoE can still deduce the optimal sample placement. We would like to leave the combination as our future work.

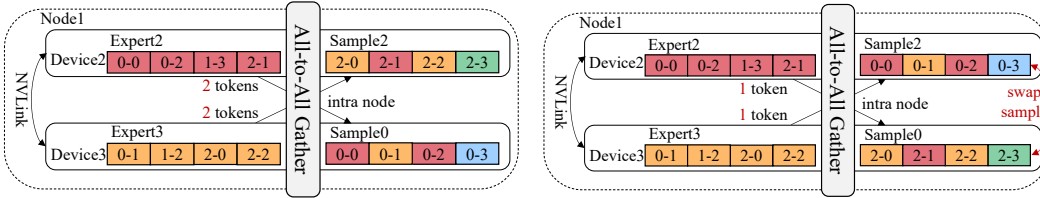

(a) Sample placement after the first stage.        (b) Sample placement after the second stage.

Figure 4: An example of the second stage optimization. Fig. 4(a) shows the MoE layer in $Node_1$ after the first stage optimization in Fig. 2(c). By applying the second stage optimization within the node, the intra-node communication can be reduced by 1 token (by swapping $sample_0$ and $sample_2$), as shown in Fig. 4(b).

connections, the most time-consuming term is usually the inter-node one. Therefore, we propose a two-stage solving strategy: the first stage optimizes $t_{inter}$ at the global scale, while the second stage minimizes $t_{intra}$ within each node, without affecting $t_{inter}$. Formally, suppose there are $N$ nodes and each node consists of $J/N$ devices, then the optimization formula of the first stage can be written as the following integer linear programming (ILP) problem:

$$\underset{\texttt{Node(SmpDev}(i))\in[\![N]\!]\text{ for }i\in[\![I]\!]}{\arg\min} \quad t_{inter}^{(l,gather)} + t_{inter}^{(l+1,scatter)}$$

$$\text{s.t.} \quad \sum_{i\in[\![I]\!]} \mathbb{I}[\texttt{Node(SmpDev}(i)) = n] = I/N \text{ for } n \in [\![N]\!] \tag{6}$$

The constraint of balance across devices in Eq. 5 is turned into the balance across nodes since we focus on inter-node communication in the first stage. After obtaining the optimal solution of the first stage, the second stage considers rearranging the samples within each node individually. For the $n$-th node, denote $[\![I]\!]_n^* \subseteq [\![I]\!]$ as the set of samples appointed to it after solving Eq. 6. And let $[\![J]\!]_n = \{j | j \in [\![J]\!] \wedge \texttt{Node}(j) = n\}$ be the set of experts reside on it ($[\![J]\!]_n$ is determined by the device placement rather than obtained by Eq. 6). Then, to optimize for the $n$-th node, we should solve the following ILP problem:

$$\underset{\texttt{SmpDev}(i)\in[\![J]\!]_n\text{ for }i\in[\![I]\!]_n^*}{\arg\min} \quad t_{intra}^{(l,gather)} + t_{intra}^{(l+1,scatter)} \quad \text{s.t.} \sum_{i\in[\![I]\!]_n^*} \mathbb{I}[\texttt{SmpDev}(i) = j] = I/J \text{ for } j \in [\![J]\!]_n \tag{7}$$

Specifically, Fig. 2 can be regarded as the optimization of the first stage, while Fig. 4 demonstrates the second stage of optimization built upon it. Although the second stage consists of $N$ ILP problems, each for one node, they are independent and can be solved concurrently.

**Polynomial-time Solver:** By dissecting the original combinatorial optimization problem, we obtain $N + 1$ ILP problems, which can be solved via existing libraries like PuLP (Mitchell et al., 2011). However, recall that we need to solve these problems for each layer in each training iteration, the efficiency of problem-solving is vital. Unfortunately, since ILP problems are NP-hard, when we try to solve them via PuLP, the time cost of solving exceeds the time cost of scatter communication and experts' computation (as evaluated in §4.4), making it impractical. Given the fact that each sample must be assigned to one device, we reconsider the ILP problems as assignment problems by transforming them into weighted bipartite matching problems, and subsequently develop a polynomial-time solver based on the widely used Kuhn-Munkres (KM) algorithm.

We first introduce how to transform the ILP problems into weighted bipartite matching problems. Let $c_{i,n}$ and $c'_{i,j}$ represent the inter- and intra-node communication volume when placing the $i$-th sample on the $j$-th device in the $n$-th node, They can be calculated using the following formulas:

$$c_{i,n} = \sum_{e\in S} num_{i,e}, \quad c'_{i,j} = \sum_{e\in S'} num_{i,e}, \quad \text{where}$$

$$S = \{e | \texttt{Node(ExpDev}(e)) \neq n\}, S' = \{e | \texttt{Node(ExpDev}(e)) = \texttt{Node}(j) \wedge \texttt{ExpDev}(e) \neq j\}. \tag{8}$$

To make the expression clearer, let $p_{i,n}, p'_{i,j} \in \{0, 1\}$ indicate whether the $i$-th sample is placed on the $n$-th node and the $j$-th device, respectively. Then, the optimization objective can be expressed as

$$t_{inter} = \alpha_{inter} + \beta_{inter} \sum_{i\in[\![I]\!],n\in[\![N]\!]} c_{i,n}p_{i,n}, \quad t_{intra} = \alpha_{intra} + \beta_{intra} \sum_{i\in[\![I]\!],j\in[\![J]\!]} c'_{i,j}p'_{i,j}, \quad \text{where}$$

$$p_{i,n} = \mathbb{I}[\texttt{Node(SmpDev}(i)) = n], \quad p'_{i,j} = \mathbb{I}[\texttt{SmpDev}(i) = j] \quad \text{for } i \in [\![I]\!], n \in [\![N]\!], j \in [\![J]\!] \tag{9}$$

---

**Algorithm 1** NetMoE Optimization

---

1: **function** `Solve`($num$)
2:     Get $c, c'$ via Eq. 8 and build bipartite graphs
3:     Get the optimal solution $p^*$ via the Kuhn-Munkres (KM) algorithm
4:     **return** the optimal sample placement according to $p^*$
5: **The Main Training Process:**
6: **for** submodule **in** model **do**
7:    **if** submodule **is** a MoE layer **then**
8:       Get $route$ from the gating network and calculate $num$ via Eq. 2
9:       Invoke `Solve`($num$) in a background thread           ▷ Offloading solving process
10:      Get $input$ from All-to-All scatter
11:      Get $output$ from expert computation
12:      $output = input + output$                                ▷ Expert residual inlining
13:      Get the optimal sample placement from the background thread
14:      Perform All-to-All gather with the optimal sample placement
15:    **else**
16:      submodule.forward()

---

After modifying the corresponding constraints, we transform the ILP problems into (0,1)-ILP problems. For instance, below presents the transformed problem for the first stage[2] (i.e., Eq. 6):

$$\underset{p_{i,n} \text{ for } i\in[\![I]\!], n\in[\![N]\!]}{\arg\min} \quad \alpha_{inter} + \beta_{inter} \sum_{i\in[\![I]\!], n\in[\![N]\!]} \left( c_{i,n}^{(l,gather)} + c_{i,n}^{(l+1,scatter)} \right) p_{i,n}$$

$$\text{s.t.} \quad \sum_{i\in[\![I]\!], n\in[\![N]\!]} p_{i,n} = I/N \text{ for } n \in [\![N]\!], \quad \sum_{n\in[\![N]\!]} p_{i,n} = 1 \text{ for } i \in [\![I]\!] \tag{10}$$

This (0,1)-ILP problem can be modeled as a weighted bipartite matching problem. In particular, consider a bipartite graph with two sets of graph nodes, $P$ and $Q$. The set $P$ represents all training samples, and $|P| = I$. The set $Q$ represents all training nodes (machines), where each training node can handle $B := I/N$ training samples. To model this, each graph node in $Q$ is duplicated $B$ times, resulting in $|Q| = I$. A weighted edge exists between every pair of graph nodes from $P$ and $Q$. Let $P_i$ represent the $i$-th training sample and $Q_n$ the $\lfloor n/B \rfloor$-th training node. The weight of the edge between $P_i$ and $Q_n$ is denoted as $W_{i,n} = c_{i,\lfloor n/B \rfloor}^{(l,gather)} + c_{i,\lfloor n/B \rfloor}^{(l+1,scatter)}$. This transformation reduces the problem of finding a minimum weight perfect matching in this bipartite graph, which can be efficiently solved to optimality in polynomial time using the Kuhn-Munkres (KM) algorithm. Fig. 5 illustrates an example of constructing a bipartite graph during the first stage in Fig. 2. The graph nodes on the left represent set $P$, and the graph nodes on the right

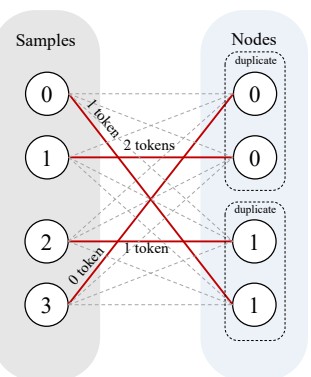

Figure 5: An example of a bipartite graph.

represent set $Q$. Each pair of graph nodes is connected by a weighted edge, depicted by a dotted line. The red edges indicate the final matching scheme, where the total weight of all matched edges is minimized.

### 3.3 Implementation

NetMoE is implemented on top of PyTorch (Paszke et al., 2019), with custom operations (e.g., the calculation of $num, c, c'$, and the KM algorithm) implemented in C++ and CUDA. The complete workflow of NetMoE is presented in Alg. 1. In addition to the problem-solving introduced in §3.2, NetMoE has been optimized in the following ways.

**Expert Residual Inlining:** In classic MoE models, residual connections are independent of the MoE layers. However, in NetMoE, the position of the training data changes after the All-to-All

---

[2]The problems of the second stage (Eq. 7) can also be transformed into (0,1)-ILP problems and solved in polynomial time similarly. We omit them here due to the space constraint and only discuss the first stage.

Table 3: Configurations of the evaluated models.

| Model Name | Base | $\frac{I}{J}$ | $S$ | $H$ | $\frac{E}{J}$ | $K$ |
|---|---|---|---|---|---|---|
| MoE-GPT-S | GPT-2 | 4 | 1024 | 768 | 2 | 2 |
| MoE-GPT-M | GPT-2 | 4 | 1024 | 1024 | 2 | 2 |
| MoE-GPT-L | GPT-2 | 4 | 1024 | 1280 | 2 | 2 |
| MoE-GPT-XL | GPT-2 | 4 | 1024 | 1600 | 2 | 2 |
| MoE-GPT-XXL | GPT-3 | 4 | 1024 | 4096 | 2 | 2 |

Table 4: Time cost of different solvers vs. the summed time cost of All-to-All scatter and expert computation in milliseconds (MoE-GPT-S, $J = 16$).

| $\frac{I}{J}$ | KM | PuLP | Scatter + Computation |
|---|---|---|---|
| 2 | 0.08 | 42.8 | 3.69 (2.63 + 1.06) |
| 4 | 0.48 | 50.1 | 7.13 (5.34 + 1.79) |
| 8 | 1.48 | 72.9 | 13.50 (10.31 + 3.19) |
| 16 | 10.82 | 143.7 | 27.31 (21.49 + 5.82) |
| 24 | 31.09 | 266.5 | 41.65 (33.82 + 7.83) |

gather operation, while the samples in the residual connections remain in their original positions. To ensure the correctness of the model, we inline the residual connections into the expert computation, as shown in line 12 of Alg. 1. This optimization ensures consistency in model accuracy before and after applying the algorithm. More details about inlining is elaborated in Appendix A.1.

**Offloading Solving Process:** The KM algorithm is hard to parallelize, making it unsuitable for highly parallelized accelerators like GPUs, so we perform the solving process on the CPU. As shown in line 9 of Alg. 1, after obtaining the routing results for the current layer, each device calculates and transfers $num$ to the CPU memory. The routing results for the next layer, required by the optimization algorithm, can be predicted by directly passing the current layer's input to the router of the next layer (Eliseev & Mazur, 2023; Tang et al., 2024). The solving process only needs to provide the new sample positions before the All-to-All gather operation. In this way, the solving process can be overlapped with the All-to-All scatter and expert computation. As we will show in §4.4, the solving time is fully hidden and thus introduces zero overhead. More discussion of algorithm selection and overlap potential is described in Appendix A.2.

## 4 EXPERIMENTS

### 4.1 EXPERIMENTAL SETUPS

We compare NetMoE with state-of-the-art methods based on dynamic expert placement, including FasterMoE (He et al., 2022) and SmartMoE (Zhai et al., 2023). We also included FastMoE (He et al., 2021) to represent a baseline without adjusting the placement of experts or samples. All experiments are conducted on a cluster consisting of 4 nodes, each equipped with 8 NVIDIA A800-SXM4-40GB GPUs. As listed in Table 2, the GPUs within each node are connected via NVLink with a 400 GB/s bandwidth, while the nodes are interconnected via InfiniBand with a 100 GB/s bandwidth. The configurations of the evaluated models are listed in Table 3. We select the GPT model architecture (Radford et al., 2019; Brown et al., 2020) as the backbone and replace all FFN layers in each model with MoE layers. In particular, since SmartMoE requires at least 2 experts on each device, we set the number of experts as $E = 2 \times J$, where $J$ is the number of GPUs in the corresponding experiment, and we fix the number of selected experts for each token as $K = 2$. By default, we utilize 8 GPUs per node to carry out the experiments, and we present the results for scenarios with fewer GPUs per node in Appendix B. All results are averaged over 50 iterations.

### 4.2 END TO END PERFORMANCE

As shown in Fig. 6, NetMoE demonstrates up to a $1.67\times$ speedup over FastMoE, a $1.37\times$ speedup over FasterMoE, and a $1.33\times$ speedup over SmartMoE. FasterMoE achieves significant optimization by overlapping expert computation and supporting dynamic expert placement. However, as the model's hidden dimension increases, the cost of communicating with experts rises, making it difficult for it to maintain the same level of acceleration. This leads to a performance gap between FasterMoE and NetMoE. On the other hand, SmartMoE outperforms FasterMoE, which is expected since SmartMoE adjusts expert placement to ensure load balancing on top of FasterMoE's optimizations. However, SmartMoE primarily focuses on balancing the computational load, without emphasizing communication efficiency. When communication becomes the primary bottleneck, the benefits of load balancing are less pronounced. Consequently, by dynamically adjusting the sample placement, NetMoE consistently outperforms the state-of-the-art systems. Last but not least,

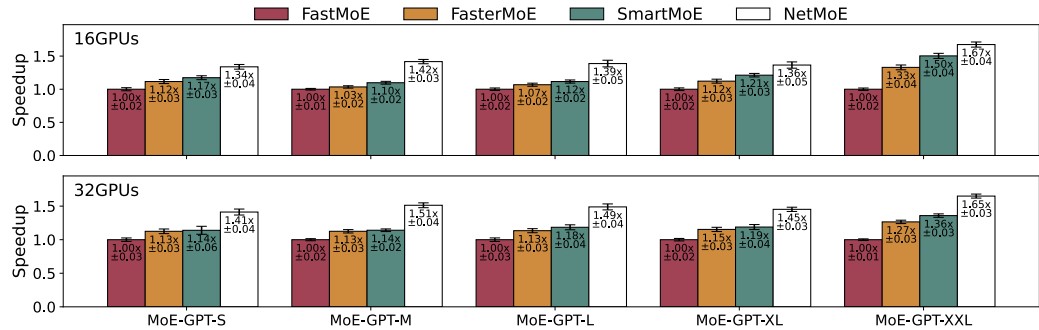

Figure 6: End-to-end speedup (mean and standard deviation) of different methods.

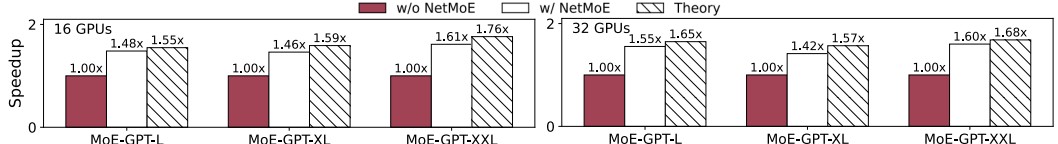

Figure 7: The actual and theoretic speedup in terms of All-to-All communication cost.

it is noteworthy that our method is compatible with dynamic expert placement. By adjusting the ExpDev(·) that is fed to our solver, NetMoE can be combined with dynamic expert placement to achieve even higher efficiency. We plan to explore this integration in our future work.

### 4.3 All-to-All Performance

As shown in Fig. 7, we conducted experiments on three kinds of model to observe the differences in All-to-All communication before and after applying NetMoE and compared these results with the theoretical optimization values provided by the solver. It can be seen that the actual speedup in All-to-All communication is slightly less than the theoretical values. This discrepancy is reasonable, as our modeling of All-to-All communication assumes ideal conditions and does not account for potential routing conflicts or hardware-induced errors. In Appendix C, we have provided more experimental results to analyze the acceleration of All-to-All communication.

### 4.4 Solver Performance

To verify the efficiency of the solver, we compared the solving time under different scales with the summed time cost of All-to-All scatter and expert computation, as shown in Table 4. KM represents the algorithm used in NetMoE, while PuLP (Mitchell et al., 2011) refers to the commonly used toolkit for solving linear programming problems. It can be observed that although the solving time exhibits super-linear growth with the increase in $I$, the solving process is consistently hidden by the All-to-All scatter and expert computation for various scenarios. In contrast, PuLP's solving time is difficult to get overlapped. This highlights the necessity of designing specialized optimization methods in scenarios with high real-time performance demands.

### 5 Conclusion

We proposed NetMoE to optimize All-to-All communication, which is the primary bottleneck in training MoE models. By leveraging data and network locality, our method dynamically adjusts the placement of training samples during training, transforming inter-node communication into intra-node communication to enhance All-to-All communication efficiency. We modeled the All-to-All communication time and the sample placement as an optimization problem and designed a polynomial-time approach to solve it. Empirical results demonstrate that NetMoE outperforms existing MoE training systems by up to $1.67\times$ in terms of training efficiency.

ACKNOWLEDGEMENT

This work is supported by National Science and Technology Major Project (2022ZD0116315), National Natural Science Foundation of China (U22B2037, U23B2048, 62402011), Beijing Municipal Science and Technology Project (Z231100010323002), China National Postdoctoral Program for Innovative Talents (BX20230012), China Postdoctoral Science Foundation (2024M750103), Beijing Natural Science Foundation (4244080), the Fund of Kunpeng and Ascend Center of Excellence (Peking University), and High-performance Computing Platform of Peking University. Bin Cui is the corresponding author.

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

## A  MORE DETAILS OF IMPLEMENTATION

### A.1  DETAIL OF EXPERT RESIDUAL INLINING

As shown in Fig. 8, the original residual addition method adds the attention output to the result obtained from the gather operation. In NetMoE, however, it is added after the scatter operation but before the gather operation. Such an inlining facilitates the adjustment of sample placement, and meanwhile ensures the correctness of computation.

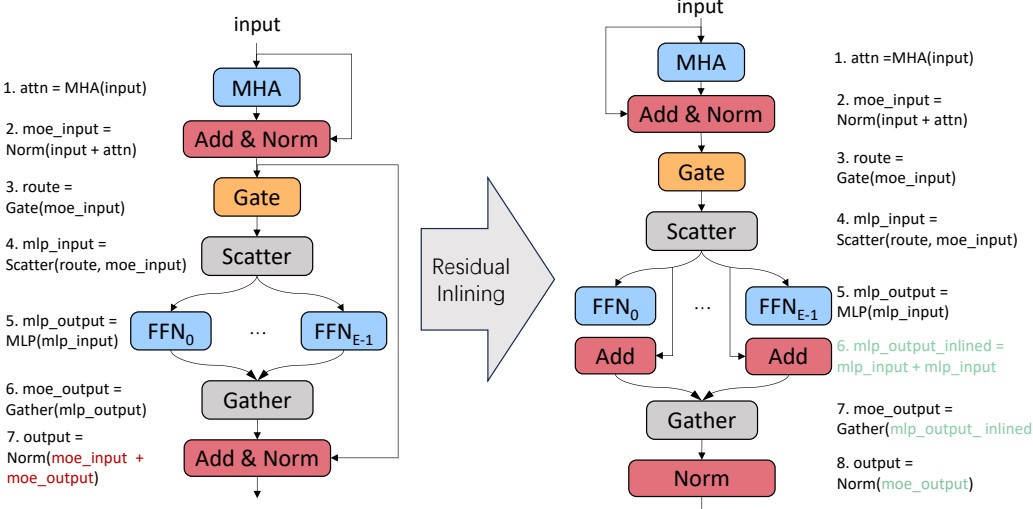

Figure 8: Illustration of the Transformer layer with and without the expert residual inlining.

### A.2  DISCUSSION OF ALGORITHM SELECTION AND OVERLAP POTENTIAL

Our design adopts the KM algorithm based on two practical factors: (1) Although the time complexity of the KM algorithm is $\mathcal{O}(I^3)$, the current training process commonly employs gradient accumulation (Tensorflow, 2019; Pytorch, 2019) due to the limited GPU memory. Thus, the value of $I$ is typically confined to an acceptable size, ensuring that the solving time can be effectively overlapped; (2) The algorithm's runtime is fully overlapped with communication phases, rendering further acceleration unnecessary for hiding the overhead of solver. While faster approximate solvers exist (Orlin & Ahuja, 1992; Duan & Pettie, 2014), their benefits would be marginal in current training configurations where computation-communication overlap already masks the optimization time.

## B  END TO END PERFORMANCE WITH FEWER GPUS PER NODE

Fig. 9 illustrates the end-to-end speedup for configurations with 2 GPUs or 4 GPUs per node. The results demonstrate that NetMoE achieves the best performance across various experimental settings, which are consistent with the results obtained when there are 8 GPUs per node (as demonstrated in Fig. 6).

It is worth noting that standard server configurations typically accommodate up to 8 NVIDIA GPUs per node. Thus, 8 GPUs per node represent a standard setup for distributed training of large language models Dubey et al. (2024); Adler et al. (2024); Dai et al. (2024); Scao et al. (2022). Although superpods like the NVIDIA GB200 NVL72 support high-speed connections (e.g., NVLink) among more than 8 GPUs, they rely on custom hardware and are prohibitively expensive. Training scenarios on superpods are rare and significantly differ from the typical scenarios in GPU clusters or clouds. Therefore, this paper opts for experiments with configurations of up to 8 GPUs per node.

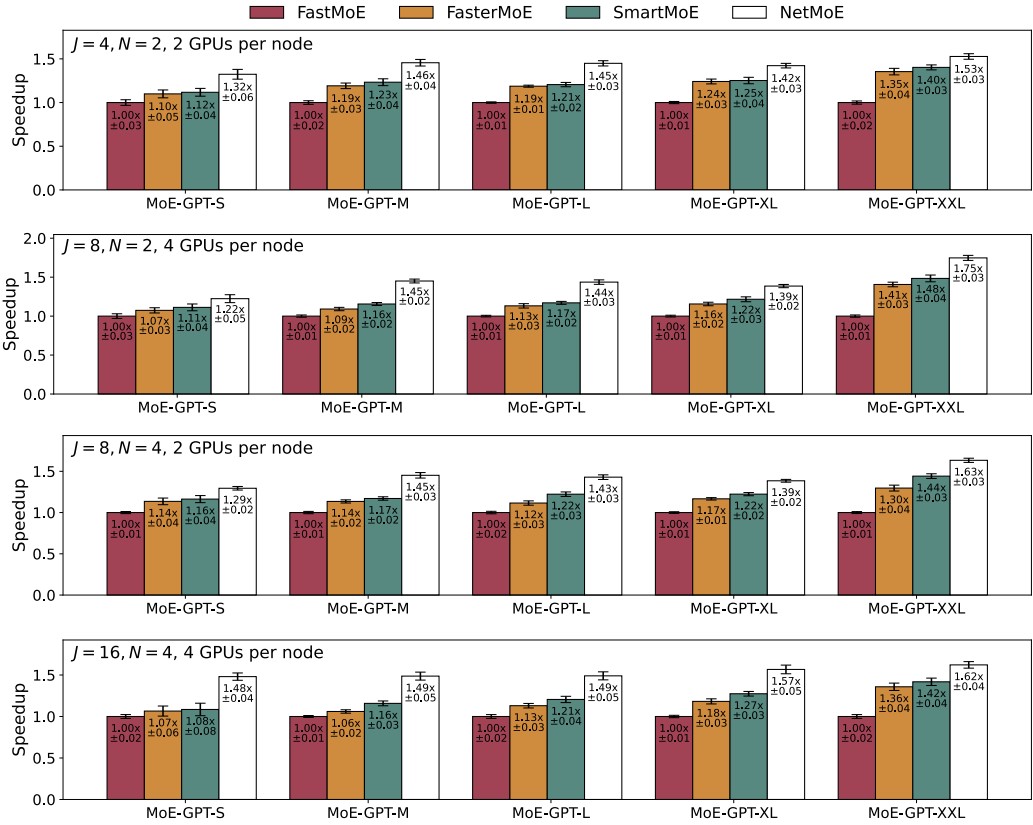

Figure 9: End-to-end speedup (mean and standard deviation) of different numbers of total devices (denoted as $J$) and numbers of nodes (denoted as $N$).

## C    DETAILED ANALYSIS OF ALL-TO-ALL COMMUNICATION OPTIMIZATION

To gain deeper insight into the source of NetMoE's optimization, we assess two kinds of statistics:

- The proportions of training samples that are exchanged across nodes or across devices by NetMoE, respectively. A higher proportion indicates more samples are adjusted across nodes/devices.
- The intra-node and inter-node communication volumes before and after applying NetMoE.

Firstly, Table 5 summarizes the mean and standard deviation across all iterations. After applying NetMoE, a great proportion of training samples are exchanged across nodes, leading to the reduction in the inter-node communication volume. It is noteworthy that although the intra-node communication volume accounts for a large proportion (i.e., $s_{intra}$ or $\frac{s_{intra}}{s_{intra}+s_{inter}}$ increases) after applying NetMoE, it will not become the performance bottleneck since the inter-node communication bandwidth is much lower. As a result, the All-to-All communication can be accelerated due to the reduction in inter-node communication volume brought by sample placement adjustment.

Secondly, since the routing result dynamically changes during the training of MoE models, to discover the impact of routing distribution, in Fig. 10 we plot (1) the reduction in inter-node communication, and (2) the proportion of samples exchanged across nodes, across different iterations. Meanwhile, we follow prior works He et al. (2022); Nie et al. (2023) to record the distribution of expert selection across different iterations in order to describe the routing distribution. It can be observed that the routing distribution changes during the model training process. However, NetMoE consistently reduces the inter-node communication by adjusting the sample placement given the dynamic distributions. Consequently, the effectiveness of NetMoE is robust to the routing distribution.

Table 5: Summary of communication volume and proportion of sequence adjustment. For communication volume, we provide the intra-node and inter-node communication volumes before and after applying NetMoE, with the increase or reduction given in parentheses. For the proportion of sequence adjustment, "Across Nodes" indicates the proportion of sequences that are exchanged across nodes, and "All" indicates the proportion of all sequences that are adjusted.

(a) 2 nodes, 16 GPUs

| | Communication Volume (MB) | | | | Proportion of Sequence Adjustment (%) | |
| | w/o NetMoE | | w/ NetMoE | | Across Nodes | All |
| | $s_{\text{intra}}$ | $s_{\text{inter}}$ | $s_{\text{intra}}$ | $s_{\text{inter}}$ | | |
|---|---|---|---|---|---|---|
| MoE-GPT-S | 168.45 ± 5.43 | 191.07 ± 5.43 | 162.24 ± 11.69 (↓ 3.69%) | 116.37 ± 11.69 (↓ 39.10%) | 43.663 ± 3.560 | 91.394 ± 1.319 |
| MoE-GPT-M | 222.89 ± 5.72 | 258.20 ± 5.72 | 214.31 ± 7.42 (↓ 3.85%) | 147.33 ± 7.42 (↓ 42.94%) | 44.455 ± 2.122 | 91.929 ± 1.163 |
| MoE-GPT-L | 281.81 ± 5.18 | 318.56 ± 5.18 | 236.53 ± 8.81 (↓ 16.07%) | 208.69 ± 8.81 (↓ 34.49%) | 42.801 ± 2.232 | 91.799 ± 1.080 |
| MoE-GPT-XL | 347.44 ± 5.68 | 402.06 ± 5.68 | 313.00 ± 8.30 (↓ 9.91%) | 256.94 ± 8.30 (↓ 36.10%) | 43.619 ± 2.267 | 91.681 ± 1.369 |
| MoE-GPT-XXL | 922.40 ± 4.16 | 989.60 ± 4.16 | 872.00 ± 8.29 (↓ 5.46%) | 570.40 ± 8.29 (↓ 42.36%) | 45.688 ± 3.006 | 92.469 ± 1.294 |

(b) 4 nodes, 32 GPUs

| | Communication Volume (MB) | | | | Proportion of Sequence Adjustment (%) | |
| | w/o NetMoE | | w/ NetMoE | | Across Nodes | All |
| | $s_{\text{intra}}$ | $s_{\text{inter}}$ | $s_{\text{intra}}$ | $s_{\text{inter}}$ | | |
|---|---|---|---|---|---|---|
| MoE-GPT-S | 167.07 ± 7.52 | 575.88 ± 7.52 | 219.21 ± 10.30 (↑ 31.21%) | 351.15 ± 10.30 (↓ 39.02%) | 72.427 ± 1.361 | 96.427 ± 0.544 |
| MoE-GPT-M | 224.24 ± 6.82 | 766.87 ± 6.82 | 288.58 ± 13.00 (↑ 28.70%) | 492.44 ± 13.00 (↓ 35.79%) | 72.340 ± 1.094 | 96.122 ± 0.461 |
| MoE-GPT-L | 280.56 ± 6.74 | 958.44 ± 6.74 | 376.66 ± 10.47 (↑ 34.25%) | 591.72 ± 10.47 (↓ 38.26%) | 72.693 ± 1.249 | 96.292 ± 0.590 |
| MoE-GPT-XL | 350.62 ± 7.10 | 1199.12 ± 7.10 | 423.37 ± 11.69 (↑ 20.75%) | 791.19 ± 11.69 (↓ 34.02%) | 72.217 ± 1.499 | 96.159 ± 0.569 |
| MoE-GPT-XXL | 884.80 ± 6.13 | 3080.00 ± 6.13 | 1201.60 ± 8.85 (↑ 35.81%) | 1884.00 ± 8.85 (↓ 38.83%) | 72.305 ± 1.886 | 96.391 ± 0.661 |

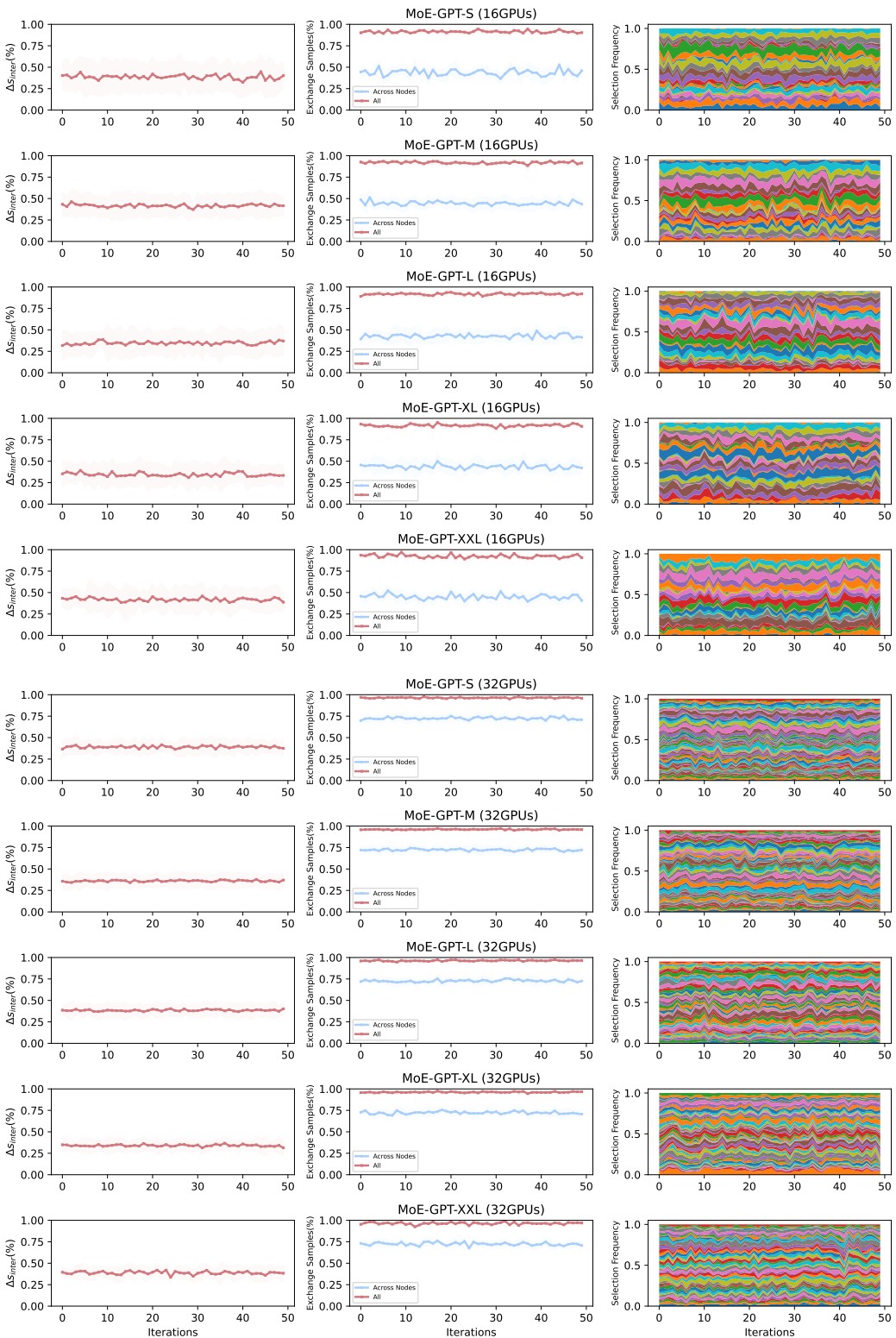

Figure 10: Left: The reduction in inter-node communication volume. Middle: The proportion of samples exchanged across nodes. Right: The distribution of expert selection (layer 0).

