# OpenReview forum: "NetMoE: Accelerating MoE Training through Dynamic Sample Placement"
_ICLR.cc/2025/Conference — ICLR 2025 Spotlight_

### Official Review · Reviewer_RL6X · 2024-10-29

**Soundness:** 3
**Presentation:** 3
**Contribution:** 3
**Rating:** 6
**Confidence:** 4

**Summary:**

Communication efficiency is a significant challenge for training efficiency in distributed Mixture of Experts (MoE) models. Unlike other papers that address this issue from a model perspective, this paper offers a solution from a data perspective. It introduces NetMoE, a method that reassigns data samples to different nodes based on locality to minimize all-to-all communication costs. The problem is formulated as an integer programming problem, and the authors derive a polynomial-time solution. Experimental results further validate the effectiveness of their approach.

**Strengths:**

1. The author demonstrates strong writing skills, clearly stating the problem and solution. The system diagram is also very clear.
2. They offer a new perspective on communication efficiency in distributed MoE by exploring how data placement can impact efficiency.
3. Experiments are provided to validate the effectiveness of their approach.

**Weaknesses:**

1. The motivation is not clearly articulated. In the motivation section, the authors mention that previous works focus on the model perspective and do not explore the data perspective, which does not convey the true motivation. Instead, it should emphasize that in certain scenarios, the model perspective may be insufficient, while a data-focused approach can achieve better efficiency.

2. The problem formulation and subsequent assumptions appear contradictory and I suspect the effectiveness of method. In Equation (1), the communication cost is defined as the maximum of intra-node and inter-node costs. However, in Section 3.2, the authors assume the maximum is the inter-node cost and address it first. This raises questions for the reviewer: if the inter-node assignment is fixed but minimizing intra-node communication results in a higher total cost than inter-node, this may lead an undesirable solution.

3. The authors transform this problem into a weighted bipartite matching problem and solve it using the Kuhn-Munkres (KM) algorithm. However, based on the reviewer's knowledge, KM is sensitive to the sample input and has a time complexity of
O(N^3) which may not be ideal for large models. The authors should justify their choice of KM as the solver.

4. The experiments do not fully validate the approach. The impact of node and device count on performance is not examined. For instance, if there are very few devices in each node but many nodes overall, inter-node communication may dominate the time. Conversely, if there are numerous devices within fewer nodes, intra-node communication could become the dominant factor in training time.

**Questions:**

Same as Weaknesses.
Q1. In what scenarios would one choose a data perspective approach over a model perspective approach?

Q2. Please revise your solution to ensure it aligns with the stated assumptions.

Q3. Explain why the Kuhn-Munkres (KM) algorithm with highest time complexity is the best choice for this problem.

Q4. Conduct additional experiments to demonstrate the impact of node and device variables on performance.

---

> ### Author Response · Authors · 2024-11-22
> **Official Comment by Authors [1/2]**
>
> We wish to express our sincere gratitude for your valuable feedback and thoughtful critique. We recognize the opportunities for improvement you've identified and believe that your insights will guide significant enhancements to our work.
>
> ***
>
> ### Weakness 1 & Question 1
>
> Indeed, our work does not provide advantageous scenarios for the model perspective and the data perspective, which could improve our work. However, we believe it does not harm the significance of our work. To the best of our knowledge, NetMoE is the first approach to accelerate MoE model training from the data perspective, and our evaluation also validates the effectiveness of NetMoE by comparing it with the approaches based on dynamic expert placement. Consequently, our work is of great significance as it offers a completely new paradigm for accelerating large language model training.
>
>
>
> Moreover, as noted in Footnote 1 on page 6 of our manuscript, our work can be integrated with the dynamic expert placement technique, combining the two perspectives together to achieve further acceleration. We would like to leave it as our future work.
>
> ### Weakness 2 & Question 2
>
> Our assumption (inter-node communication cost is higher than intra-node communication cost) is reasonable: as shown in Table 2, the intra-node bandwidth is 4 times that of inter-node bandwidth. Intuitively speaking, as long as the intra-node communication volume (denoted as $ s_{intra} $ in Section 3 of our work) is less than 4 times that of the inter-node communication volume (denoted as $ s_{inter} $), the inter-node communication dominates. To address the reviewer's concern, we measure the intra-node and inter-node communication volumes (i.e., $ s_{intra},s_{inter} $) before and after applying NetMoE. The results are presented in Table 5 of Appendix C in the revised manuscript.
>
>
>
> The results demonstrate that the inter-node communication volume reduces substantially after applying NetMoE. Although the intra-node communication volume accounts for a large proportion (i.e., $ s_{intra} $ or $ \frac{s_{intra}}{s_{intra} + s_{inter}} $ increases) after applying NetMoE, it will not become the performance bottleneck since the ratio $ \frac{s_{intra}}{s_{inter}} $ remains significantly smaller than 4. This indicates that the All-to-All communication can be accelerated due to the reduction in inter-node communication volume. Consequently, it is feasible to minimize the inter-node communication first, and then minimize the intra-node communication with a fixed inter-node communication.
>
>
>
> In fact, due to the difference in bandwidth, it is a common practice to prioritize the reduction in the inter-node communication. For instance, [1] also focuses on minimizing the inter-node communication volume rather than the intra-node one (detailed in Section 3.1 of their paper). As a result, we believe the two-stage problem-solving in our work is reasonable and practical.
>
>
>
> [1] Zhuang et al. On Optimizing the Communication of Model Parallelism. https://arxiv.org/abs/2211.05322.

---

> > ### Comment · Reviewer_RL6X · 2024-11-22
> > **Please explain more about following scenario**
> >
> > Thanks for your explanation. However, I think the ratio of intra-node and inter-node communication volume depends on how many nodes you have. If you only have a few nodes and each node contains massive devices,  I assume it induces much more intra-node communication cost than inter-node. In this case, your assumption (inter-node communication cost is higher than intra-node communication cost) is not valid. Can you explain how your method works in this scenario?

---

> > > ### Author Response · Authors · 2024-11-22
> > >
> > > Thank you for your prompt response and for the insightful feedback. We agree that the intra-node communication may dominate when we only have a few nodes and each node contains massive devices. In such cases, our work may not work well as our assumption does not hold true.
> > >
> > > Nevertheless, we would like to explain that, typically, there are at most 8 NVIDIA GPUs per node in standard server configurations (which is far from "massive"). Thus, as mentioned in our previous response, 8 GPUs per node represents a typical configuration in distributed training of large language models. Notable examples include:
> > >
> > > + Meta trained the Llama 3 405B model with up to 16K GPUs, with server configurations detailed on page 9 of their paper [1]:
> > >
> > > > Llama 3 405B is trained on up to 16K H100 GPUs... Each server is equipped with eight GPUs and two CPUs. Within a server, the eight GPUs are connected via NVLink.
> > >
> > > + NVIDIA trained the Nemotron-4-340B-Base model with 768x8 GPUs, as detailed on page 4 of their technical report [2]:
> > >
> > > > Nemotron-4-340B-Base was trained using 768 DGX H100 nodes; each node contains 8 H100 80GB SXM5 GPUs based on the NVIDIA Hopper architecture.
> > >
> > > + The BLOOM 176B model was trained with 48x8 GPUs, as detailed on page 18 of the technical report [3]:
> > >
> > > > Training was conducted on 48 nodes, each having 8 NVIDIA A100 80GB GPUs (a total of 384 GPUs)
> > >
> > > + DeepSeekMoE was trained with the configuration of 8 GPUs per node for both the A100 and H100 clusters, as detailed on page 8 of the technical report [4]:
> > >
> > > > Each node in the A100 cluster contains 8 GPUs connected pairwise via the NVLink bridge. The H800 cluster also features 8 GPUs per node, interconnected using NVLink and NVSwitch within nodes. For both A100 and H800 clusters, InfiniBand interconnects are utilized to facilitate communication across nodes.
> > >
> > >
> > >
> > > Although superpods like NVIDIA GB200 NVL72 [5] do support high-speed connection (e.g., NVLink) among more than 8 GPUs, they rely on custom hardware equipment and are exceedingly expensive. Scenarios of training on superpods are rare and significantly different from the common scenarios of training in GPU clusters or clouds.
> > >
> > >
> > >
> > > To conclude, our assumption holds true in general cases. In addition, in our revised manuscript, we have conducted experiments with varying numbers of GPUs per node to evaluate the effectiveness of NetMoE (Figure 9 of Appendix B), and provided detailed statistics to demonstrate that inter-node communication cost is still the performance bottleneck in All-to-All communication (Table 5 of Appendix C).
> > >
> > >
> > >
> > > We hope our response addresses the reviewer's concerns. And we sincerely hope that you can re-evaluate your rating of our work.
> > >
> > >
> > >
> > > [1] Meta. The Llama 3 Herd of Models. https://arxiv.org/abs/2407.21783.
> > >
> > > [2] NVIDIA. Nemotron-4 340B Technical Report. https://arxiv.org/abs/2406.11704.
> > >
> > > [3] BigScience. BLOOM: A 176B-Parameter Open-Access Multilingual Language Model. https://arxiv.org/abs/2211.05100.
> > >
> > > [4] DeepSeek. DeepSeekMoE: Towards Ultimate Expert Specialization in Mixture-of-Experts Language Models. https://arxiv.org/abs/2401.06066.
> > >
> > > [5] NVIDIA. NVIDIA GB200 NVL72. https://www.nvidia.com/en-us/data-center/gb200-nvl72.

---

> > > > ### Comment · Reviewer_RL6X · 2024-11-22
> > > >
> > > > Thanks for providing those practical scenarios and I hope you can add them in your revised papaer.

---

> > > > > ### Author Response · Authors · 2024-11-23
> > > > >
> > > > > Thank you for appreciating our work and for the time and effort you have dedicated as a reviewer! We have included this part of the discussion in Appendix B of the revised paper.

---

> ### Author Response · Authors · 2024-11-22
> **Official Comment by Authors [2/2]**
>
> ### Weakness 3 & Question 3
>
> As pointed out by the reviewer, the Kuhn-Munkres (KM) algorithm does have a cubic time complexity. However, it is much more efficient than solving the original target optimization problem (defined in Equation 6,7), and fits the scenario of large language model training well.
>
> + Firstly, as discussed in Section 3.2 and evaluated in our experiments (Table 4), the target optimization problem (defined in Equation 6,7) is an integer linear programming (ILP) problem, which is NP-hard. Although existing libraries like PuLP support solving ILP problems, it takes a long time for PuLP to accomplish the problem-solving, making it infeasible to hide the solving cost by overlapping. To cope with this problem, we designed a method based on the Kuhn-Munkres (KM) algorithm that can be done in polynomial time. Experiment results in Table 4 demonstrate that our KM-based solving algorithm is much faster than directly solving with PuLP, and we are able to completely overlap the problem-solving.
>   In addition, the KM algorithm is extremely widely used to solve assignment problems. Considering that our work focuses on how to reduce the communication cost by re-assigning the training samples among the GPU devices, the KM algorithm is suitable for our work.
> + Secondly, the complexity of KM algorithm is related to the batch size per device during training (yet independent of model size). In distributed training of large language models, due to the constraint of GPU memory, it is infeasible to support a large batch size per device at once. Instead, it is common to leverage the gradient accumulation technique [2,3] to ensure that the batch size per device for each gradient accumulation step is small (while allowing the model to be updated with the gradients of a large batch). Thus, the batch size per device for each step is typically small.
>   In Section 4.4 of the revised manuscript, we further conduct an experiment with a batch size per device of 24 (a higher value would lead to out-of-memory errors) to examine the effectiveness of our KM-based solving algorithm. The results show that the problem-solving can still be overlapped well.
>
>
>
> In the revised manuscript, we have added more discussion about why we solve the problem via KM algorithm in Section 3.2 and provided the experimental results in Section 4.4.
>
>
> [2] Pytorch, “Gradient accumulation pytorch,” https://gist.github.com/thomwolf/ac7a7da6b1888c2eeac8ac8b9b05d3d3.
>
> [3] Tensorflow, “Gradient accumulation tensorflow,” https://github.com/tensorflow/tensorflow/pull/32576.
>
>
>
> ### Weakness 4 & Question 4
>
> In our original manuscript, we consider training over 2 and 4 nodes, with each node consisting of 8 GPUs, which is a typical configuration in distributed training of large language models. To address the reviewer's concern, we further conduct experiments with 2 and 4 GPUs per node, respectively. The results are provided in Appendix B of the revised manuscript. Overall, NetMoE still consistently achieves the best performance.

---

> > ### Comment · Reviewer_RL6X · 2024-11-22
> >
> > Thanks for providing more experiments and discussion of KM algorithm.

---

### Official Review · Reviewer_JYL4 · 2024-11-03

**Soundness:** 4
**Presentation:** 3
**Contribution:** 3
**Rating:** 8
**Confidence:** 3

**Summary:**

The paper proposes a topology-aware sample placement scheduling approach to optimize All-to-All communication in MoE training.

**Strengths:**

•	Theoretical Rigor: This paper is thorough in formulating the communication challenges and solution as an optimization problem, with clear problem modeling and a detailed, polynomial-time solution.

•	Practicality: The method can integrate with existing MoE training systems while enhancing training efficiency.

•	Empirical Validation: Experimental results across various configurations validate NetMoE’s improvements in All-to-All communication and overall training efficiency.

**Weaknesses:**

•	Experimental Context: The paper could benefit from a more comprehensive discussion on the "data locality" conditions required to achieve the claimed speedups in real-world setups. Also, details on the distribution of data locality across real-world training tasks (and the one used in experiment) would give more insight into NetMoE's practical performance.

•	Discussion on Experiment Setup: Given that inter-node expert parallelism can incur heavy communication costs, it would help if the authors provided reasoning for prioritizing inter-node expert parallelism over potentially less intensive techniques like a hybrid one: intra-node expert parallelism + inter-node pipeline parallelism.

•	More Baseline Comparisons: Additional baselines, particularly concerning dynamic expert placement, would highlight NetMoE’s comparative advantages and limitations.

**Questions:**

•	How does the data locality used in the experiment compare to typical training scenarios, and what impact might this have on expected performance?

•	Why is inter-node expert parallelism favored over pipeline or other model parallelism techniques in this context?

•	Is an auxiliary loss mechanism incorporated to mitigate expert selection skew, and if so, does it affect the performance of NetMoE?

---

> ### Author Response · Authors · 2024-11-22
>
> We deeply appreciate your acknowledgment of our work and your constructive feedback. We are confident that our work will be significantly improved by incorporating your insights.
>
> ***
>
> ### Weakness 1 & Question 1
>
> The data locality is a well-known characteristic in MoE models and has been motivating many works to accelerate MoE training or inference [1,2,3,4,5]. Although some studies have tried to investigate the data locality given specific pre-trained MoE models and datasets [6,7], we wish to clarify that given any dataset and MoE model, the routing distribution would dynamically change during the training, so it is hard to control the distribution to examine the speedup in end-to-end model training. Therefore, to address the reviewer's comment, we assess whether NetMoE can reduce the inter-node communication volume when facing the dynamicity in routing distributions.
>
>
>
> To be specific, we follow prior works [1,2] to record the distribution of expert selection across different iterations in order to describe the routing distribution and record the reduction in inter-node communication as well. The results are provided in Figure 10 of Appendix C in the revised manuscript. It can be seen that the routing distribution changes during the model training process, while NetMoE consistently reduces the inter-node communication by adjusting the sample placement given the dynamic distributions. Consequently, the effectiveness of NetMoE is general to various data locality conditions.
>
>
>
> [1] He et al. FasterMoE: modeling and optimizing training of large-scale dynamic pre-trained models. https://dl.acm.org/doi/10.1145/3503221.3508418.
>
> [2] Nie et al. FlexMoE: Scaling Large-scale Sparse Pre-trained Model Training via Dynamic Device Placement. https://arxiv.org/abs/2304.03946.
>
> [3] Zhai et al. SmartMoE: Efficiently Training Sparsely-Activated Models through Combining Offline and Online Parallelization. https://www.usenix.org/system/files/atc23-zhai.pdf.
>
> [4] Li et al. Accelerating distributed MoE training and inference with lina. https://www.usenix.org/system/files/atc23-li-jiamin.pdf.
>
> [5] Yao et al. Exploiting inter-layer expert affinity for accelerating mixture-of-experts model inference. https://arxiv.org/pdf/2401.08383.
>
> [6] Jiang et al. Mixtral of experts. https://arxiv.org/abs/2401.04088.
>
> [7] Xue et al. Openmoe: An early effort on open mixture-of-experts language models. https://arxiv.org/abs/2402.01739.
>
>
>
> ### Weakness 2 & Question 2
>
> Expert parallelism, tensor parallelism, data parallelism, and pipeline parallelism can be combined to achieve hybrid parallel training of large language models. Given the fact that tensor parallelism is usually applied within nodes due to its high communication volume [8], if we wish to avoid inter-node expert parallelism, there must be $ TP \times EP \leq 8 $, where $ TP $ and $ EP $ are the parallel degrees of expert parallelism and tensor parallelism, respectively. Undoubtedly, this would lead to a limited parallel configuration space. As a result, our work considers a more general case where expert parallelism involves both intra-node and inter-node communication for the experiments.
>
>
>
> [8] Singh et al. A Hybrid Tensor-Expert-Data Parallelism Approach to Optimize Mixture-of-Experts Training. https://arxiv.org/abs/2303.06318.
>
>
>
> ### Weakness 3 & Question 3
>
> In our experiments, we have compared NetMoE with two state-of-the-art baselines that are based on dynamic expert placement, which are the FasterMoE and SmartMoE. The experimental results demonstrate that NetMoE consistently outperforms the baselines in terms of training efficiency. Besides, we did not incorporate any auxiliary loss in our experiments.

---

> > ### Comment · Reviewer_JYL4 · 2024-11-25
> >
> > Thanks for your explanation and I have no more questions.

---

> > > ### Author Response · Authors · 2024-11-25
> > >
> > > Thank you for your thorough review and feedback. We're glad to have addressed your questions, and we appreciate your valuable insights.

---

### Official Review · Reviewer_rdXW · 2024-11-05

**Soundness:** 3
**Presentation:** 3
**Contribution:** 3
**Rating:** 6
**Confidence:** 4

**Summary:**

This paper proposes to use a dynamic sample placement to speed up the MoE training. Specifically, this paper adopts a mathematical model to simulate the number of inter-node communication and intra-node communication and solve the integer programming problem to figure out the best sample allocation of the sample to reduce inter-node communication inspired by the locality in networks. This paper successfully reduces the all2all gather communication in training and achieve speed up.

**Strengths:**

1. This paper tackles the MoE training efficiency from a novel perspective, that is the data locality perspective. It dynamically locates the data to reduce the inter-node communication in all2all gathering.
2. The results shows improvements compared with baselines, signifying the effectiveness of the method.
3. The modeling of the networking problem is inspiring to the reviewer.

**Weaknesses:**

1. The scalability of the method is questionable, e.g., the improvements for 32 GPUs is smaller then the improvements for 16 GPUs. This leads to the question that what will happen if we continue increasing the number of GPUs? Will the improve converges to zero?
2. When there are more GPUs, the communication should take a larger portion in the total time? Why the method here, which primarily focuses on optimizing communication, have less significant improvements.

**Questions:**

See Weaknesses. And

Moving the sample should incurs more movements compared a subset of the tokens in the sample. Why moving sample gives less communication overhead?

---

> ### Author Response · Authors · 2024-11-22
>
> We are truly grateful for your careful review and thoughtful questions, which are valuable to our work.
>
> ***
>
> ### Weaknesses
>
> Due to the lack of GPU resources, we are not able to examine the scalability of NetMoE with more GPUs currently (e.g., 64 GPUs). However, we wish to highlight that the improvement of NetMoE does not diminish as the number of GPUs increases. To be specific, as shown in Figure 5 in the original manuscript (which is now Figure 6 in the revised manuscript), NetMoE achieves a slightly higher speedup with 32 GPUs compared to 16 GPUs for 4 of the 5 experimented models. For MoE-GPT-XXL, the speedup is very close: with 16 GPUs, the speedup is $ 1.67 \pm 0.039 $, and with 32 GPUs, it is $ 1.65 \pm 0.030 $. Across all experiments, the average speedup of NetMoE on 32 GPUs surpasses that on 16 GPUs. Consequently, the speedup delivered by NetMoE is robust to the increase in the number of GPUs.
>
>
>
> To avoid ambiguity, in the revised manuscript, we have provided the standard deviation of end-to-end speedup in Figure 6.
>
>
>
> ### Questions
>
> To compute attention efficiently, all tokens of a training sample should reside on the same GPU device. If we exchange only part of the tokens, then the tokens of each training sample would be distributed across different GPU devices. In this case, it necessitates substantial extra communication to accomplish the attention computation, which is counterproductive. Therefore, we adjust the placement at the granularity of training samples.

---

> > ### Comment · Reviewer_rdXW · 2024-11-26
> > **Response to the authors**
> >
> > Thank you for your rebuttals. This makes sense to me and I have no further questions.

---

> > > ### Author Response · Authors · 2024-11-26
> > >
> > > Thank you for your detailed review and feedback. We’re pleased to have addressed your questions and appreciate your valuable insights.

---

### Official Review · Reviewer_EW5K · 2024-11-06

**Soundness:** 3
**Presentation:** 3
**Contribution:** 3
**Rating:** 8
**Confidence:** 2

**Summary:**

The whole idea of NetMoE is that we want to reduce the All-to-All scatter & gather communications by reducing the amount of cross-node/device routing of tokens. To achieve this we will adjust the sample/sequence that would minimize the inter-node & intra-node communication volume. This is (approximately) solvable as a weighted bipartite matching / assignment problem between training samples and machines, as shown in Eqn 9 and 10.

The authors conduct experiments on GPT pretraining and compare with dynamic expert placement baselines as FasterMoE and SmartMoE. NetMoE generally has higher speedup (Figure 5) and the actual speedup is close to the theoretically optimal speedup (Figure 6).

**Strengths:**

1. The paper is well motivated and the writing is pretty clear. I have no difficulty on understanding the overall idea of sample adjustment (from Figure 2) and the optimization challenges & solutions (Equation 5, 8, 10) upon the first time of reading.

2. Clever design: reformulating the ILP to a weighted bipartite matching / assignment problem and using Hungarian algorithm that has shorter solving time than communication time (so we can have actual speedup).

**Weaknesses:**

I don't have strong opposition to the overall idea of sequence adjustments for MoE but I believe the scope and limitations should be more clearly defined:

1. The authors should provide a summary statistics on how many sequences are actually adjusted across nodes/devices during training and how it is correlated with the MoE specialization / router probability.

2. A small-scaled ablation experiment is definitely needed to show if this communication volume reduction is robust w.r.t. the choice of dataset mixtures, as the performance of NetMoE might be data dependent.

3. Table 4 is concerning because the limit of KM algorithm to use less time than all-scatter is $I/J \sim 24$ (24 is my scaling extrapolation of Table 4's $I/J = 16$ results as KM's time complexity scales cubically w.r.t. # nodes, and $(24/16)^3 * 1 > (24/16) * 2$). A batch size of 24 per device is not a sufficiently large number.

**Questions:**

1. The sequence adjustment is done per iteration and per layer and composable of reducing the all-gather communication of this layer and all-scatter of next-layer (Eqn. 7). The reduction from all-gather is clear, but I don't understand how it is even possible to reduce the all-scatter costs of *next-layer* as we even don't know what is the routing probability due to an attention block before the MoE.

2. I don't understand how does expert inline residual fix the position issues of residual stream (it might be helpful to give a diagram as line 12 in Algorithm 1 is not sufficiently clear)

---

> ### Author Response · Authors · 2024-11-22
> **Official Comment by Authors [1/2]**
>
> We sincerely appreciate your recognition of our work and thank you for your constructive suggestions. We believe that addressing your suggestions will substantially improve our work.
>
> ***
>
> ### Weakness 1
>
> To address the reviewer's concern, in Appendix C of the revised manuscript, we consider two kinds of statistics to assess the source of performance improvement of NetMoE:
>
> + The proportions of training samples that are exchanged across nodes/devices by NetMoE. A higher proportion indicates more samples are adjusted across nodes/devices.
> + The intra-node and inter-node communication volumes before and after applying NetMoE.
>
>
>
> Firstly, we summarize the mean and standard deviation across all iterations in Table 5 of Appendix C in the revised manuscript. After applying NetMoE, a great proportion of training samples are exchanged across nodes, leading to the reduction in the inter-node communication volume. It is noteworthy that although the intra-node communication volume accounts for a large proportion (i.e., $ s_{intra} $ or $ \frac{s_{intra}}{s_{intra} + s_{inter}} $ increases) after applying NetMoE, it will not become the performance bottleneck since the inter-node communication bandwidth is much lower. As a result, the All-to-All communication can be accelerated due to the reduction in inter-node communication volume brought by sample placement adjustment.
>
>
>
> Secondly, to discover the impact of router probability, in Figure 10 of Appendix C in the revised manuscript, we plot (1) the reduction in inter-node communication, and (2) the proportion of samples exchanged across nodes, across different iterations. Meanwhile, we follow prior works [1,2] to record the distribution of expert selection across different iterations in order to describe the routing distribution. It can be observed that the routing distribution changes during the model training process. However, NetMoE consistently reduces the inter-node communication by adjusting the sample placement given the dynamic distributions. Consequently, the effectiveness of NetMoE is robust to the router probability.
>
>
>
> [1] He et al. FasterMoE: modeling and optimizing training of large-scale dynamic pre-trained models. https://dl.acm.org/doi/10.1145/3503221.3508418.
>
> [2] Nie et al. FlexMoE: Scaling Large-scale Sparse Pre-trained Model Training via Dynamic Device Placement. https://arxiv.org/abs/2304.03946.
>
>
>
> ### Weakness 2
>
> We wish to clarify that for any dataset mixture, the routing distribution would be dynamic during the training of MoE models. Thus, it is more important to be robust w.r.t. routing distributions. As elaborated in our response to Weakness 1 and demonstrated in Figure 10 of Appendix C, the reduction in communication volume is consistent given various routing distributions, demonstrating the robustness of our work.
>
>
>
> ### Weakness 3
>
> In distributed training of large language models, due to the constraint of GPU memory, it is infeasible to support a large batch size per device at once. Instead, it is common to leverage the gradient accumulation technique [3,4] to ensure that the batch size per device for each gradient accumulation step is small (while allowing the model to be updated with the gradients of a large batch). Thus, the batch size per device for each step is typically small. In fact, for Table 4, when we try to increase the batch size per device to 32, the training task encounters an out-of-memory error.
>
>
>
> In addition, in the original manuscript, Table 4 only compares the time cost of the problem-solving and the scatter operation for simplicity, while in practice, the problem-solving can be overlapped as long as its time cost is smaller than the summed time cost of the scatter operation and the expert computation. To further address the reviewer's concern, we conduct an experiment with a batch size per device of 24, which gives the result that the time cost of problem-solving is 31.09ms, while the summed time cost of the scatter operation and the expert computation is 41.65ms (33.82ms + 7.83ms), showing that the problem-solving process can still be overlapped well.
>
>
>
> In the revised manuscript of our work, we have incorporated the results above in Table 4, and added more discussion about the batch size in Section 3.3.
>
>
>
> [3] Pytorch, “Gradient accumulation pytorch,” https://gist.github.com/thomwolf/ac7a7da6b1888c2eeac8ac8b9b05d3d3.
>
> [4] Tensorflow, “Gradient accumulation tensorflow,” https://github.com/tensorflow/tensorflow/pull/32576.

---

> ### Author Response · Authors · 2024-11-22
> **Official Comment by Authors [2/2]**
>
> ### Question 1
>
> We feed the current layer's input directly to the next layer's router to predict the expert routing of the next layer. Such a prediction method is also adopted in existing works [5,6]. The rationality is that, since residual connections are needed in transformer layers, the inputs to the routers of two consecutive layers should share certain similarities. This prediction doesn't need high accuracy and serves only as a guide during algorithmic optimization.
>
>
>
> [5] Eliseev and Mazur. Fast inference of mixture-of-experts language models with offloading. https://arxiv.org/abs/2312.17238.
>
> [6] Tang et al. HOBBIT: A Mixed Precision Expert Offloading System for Fast MoE Inference. https://arxiv.org/abs/2411.01433.
>
>
>
> ### Question 2
>
> In Appendix A of the revised manuscript, we have provided a detailed visual explanation of the residual inlining. Specifically, the original residual addition method adds the attention output to the result obtained from the gather operation. In NetMoE, however, it is added after the scatter operation but before the gather operation. Such an inlining facilitates the adjustment of sample placement, and meanwhile ensures the correctness of computation.

---

> ### Comment · Reviewer_EW5K · 2024-11-30
> **Response to Author's response**
>
> Thanks for the response.
>
> The new results in Appendix A, B, C are great. It addresses my first and (at least) half of my second concern. My third concern is also half addressed and I would suggest to add a brief discussion on relevant MoE training recipes that do not need large batch size per device, and when we indeed need large batch size per device (and for GPU utilization purpose we sometimes need sufficiently large *microbatch* to medium-sized models), what is the other approximate solver available (this doesn't need to be perfect, but should still be better than random).
>
> My 2 questions are well answered. Thanks for this clear response!

---

> > ### Author Response · Authors · 2024-11-30
> >
> > Thank you for your response and new suggestions. We would like to provide further clarification on your concerns:
> >
> > Firstly, the current MoE model training literature lacks documentation or technical reports specifying the local batch size used per gradient accumulation step during training, as it is constrained by memory limitations without affecting model convergence. However, we can infer the local batch size from MoE training scripts provided by open-source frameworks. In the scripts from DeepSpeed [1] and Megatron [2] used for MoE training, the maximum value is 8, indicating that most training scenarios involve a relatively small local batch size.
> >
> > Secondly, numerous optimization methods [3][4] exist for the KM algorithm. Notably, [4] supports the trade-off between efficiency and accuracy, allowing for lower time complexity if larger errors are permitted. lf a larger local batch size is indeed necessary for training, the solving method can be replaced with more efficient alternatives mentioned above, yet they may introduce certain errors to the achieved solutions.
> >
> > Last but not least, we wish to clarify that our primary contribution lies in proposing a method to optimize communication by adjusting sample placement. We selected the KM algorithm as the solver for its simplicity, as its solving time can be effectively overlapped with communication and computation. We acknowledge that the solver can be replaced with more advanced algorithms to enhance solving efficiency, and we believe our work is able to inspire follow-up works to explore diverse approaches to support more scenarios.
> >
> > We hope this response addresses your concerns. According to the timeline of the ICLR reviewing process, we cannot submit a revised manuscript at this stage. However, we are committed to including these discussions in the final version.
> >
> > [1] Megatron-Deepspeed. ds_pretrain_gpt_1.3B_MoE128.sh. https://github.com/microsoft/Megatron-DeepSpeed/blob/main/examples_deepspeed/MoE/ds_pretrain_gpt_1.3B_MoE128.sh.
> >
> > [2] Megatron. train_mixtral_8x7b_distributed.sh. https://github.com/NVIDIA/Megatron-LM/blob/core_r0.9.0/examples/mixtral/train_mixtral_8x7b_distributed.sh.
> >
> > [3] Orlin, James B.; Ahuja, Ravindra K. New scaling algorithms for the assignment and minimum mean cycle problems. https://link.springer.com/article/10.1007/BF01586040.
> >
> > [4] Duan, Ran; Pettie, Seth. Linear-Time Approximation for Maximum Weight Matching. https://web.eecs.umich.edu/~pettie/papers/ApproxMWM-JACM.pdf.

---

> > > ### Comment · Reviewer_EW5K · 2024-12-01
> > > **Thanks for the response**
> > >
> > > Thanks for the response. I would suggest add this discussion to the paper and I believe this paper belongs to the category of "good paper" in ICLR. I raise my score to "accept".

---

> > > > ### Author Response · Authors · 2024-12-01
> > > >
> > > > We sincerely appreciate your recognition of our work and the time and effort you have devoted as a reviewer! We will include discussions on this aspect in the final version of the paper.

---

### Official Review · Reviewer_v9Qs · 2024-11-06

**Soundness:** 4
**Presentation:** 3
**Contribution:** 4
**Rating:** 8
**Confidence:** 4

**Summary:**

This paper presents NetMoE, a novel framework designed to optimize the routing of samples in Mixture of Experts (MoE) models by taking into account the actual inter an intro-node communication bandwidth. The goal is to minimize the *time* the routing process takes, which usually amount to minimize inter-node expert routing in the All-to-All communications, while being mathematically equivalent to the standard routing procedure. This paper formulates the problem as an integer linear programming optimization problem, and relaxes it so that an approximate solution can be found sufficiently fast dynamically at each stage of the MoE. Experimental results demonstrate that NetMoE outperforms existing MoE training systems, achieving up to a 1.67x speedup in training efficiency.

**Strengths:**

* **The problem is clearly motivated:** The challenges of routing samples in MoE are clearly written, making the goal of this paper feel natural after reading the first two sections.
* **Challenges of ILP solving are made clear, and the proposed solution seem effective:** The building to the final approximate method is clear and well motivated through empirical results in Tab.4. The optimization gap between the optimal and the approximate solution seem reasonable in Fig.6.
* **Non negligible empirical benefits of the method are demonstrated:** The speedup brought by NetMoE compared to Dynamic Expert Placement methods seem significant in the experiments displayed.

**Weaknesses:**

* **Notations and problem formulation hard to follow:** Many notations are introduced, making the reading of section 3 a bit cumbersome. Maybe putting some of the mathematical details and ILP formulations in Appendix could help lighten the section and make it more readable?
* **No comparison with methods using a modification in the model definition:** While methods introduced in Sec. 2.2 change the convergence property of the model in terms of iterations, the fact that they allow for more iterations per time unit could counter this. Would it be possible to also compare NetMoE to these methods (e.g., in terms of *"time to reach a certain level of perplexity"*)?

**Questions:**

see Weaknesses.

**typo:** In table 1 *"number of of nodes"*.

---

> ### Author Response · Authors · 2024-11-22
>
> Thank you for your acknowledgment and insightful comments. Your feedback is extremely helpful, and we are committed to addressing each question you have raised.
>
> ****
>
> ### Weakness 1: Notations and problem formulation hard to follow.
>
> To address the reviewer's concern, in the revised manuscript, we have added a figure (Figure 3 on page 4) to present the overview of Section 3, which briefly includes the problem formulation, two-stage dissection, polynomial-time solver, and implementation of our work. We hope it can improve the readability of our work.
>
>
>
> ### Weakness 2: No comparison with methods using a modification in the model definition.
>
> We agree that the approaches based on modification in model definition (which impact model convergence) could train with more iterations to reach a similar level of perplexity. However, NetMoE is orthogonal to such lossy approaches. In other words, they can be integrated with our work to achieve better efficiency. Since large language model training is time-consuming and costly, we cannot keep training models until convergence. As a result, our evaluation focuses on lossless approaches that do not affect model convergence.
>
>
>
> In addition, we wish to highlight that developing lossless approaches is timely and important. To be specific, when applying lossy approaches, we usually need to run numerous trials to tune the hyper-parameters (e.g., we need to adjust the weight of the topology-aware routing loss [1], or need to tune the hyper-parameters for different communication channels [2]), which is impractical since each trial of large language model training may take days or even months.
>
>
>
> In Section 2.2 of the revised manuscript, we have added more discussion to address the reviewer's concern.
>
>
>
> [1] Chen et al. TA-MoE: Topology-Aware Large Scale Mixture-of-Expert Training. https://arxiv.org/abs/2302.09915.
>
> [2] Zeng and Xiong. SCoMoE: Efficient Mixtures of Experts with Structured Communication. https://openreview.net/pdf?id=s-c96mSU0u5.

---

### Meta-Review · Area_Chair_1xBt · 2024-12-18

**Metareview:**

Summary:
The paper presents NetMoE, a novel framework for optimizing communication in distributed Mixture-of-Experts (MoE) model training by taking a data-centric approach to sample placement. The key innovation is formulating the problem as an integer linear programming optimization that minimizes inter-node communication costs while maintaining model accuracy. The method achieves up to 1.67x speedup compared to existing approaches.

Main strengths:

- Novel perspective on MoE optimization by addressing communication efficiency from a data placement angle rather than model architecture
- Strong theoretical foundation with clear problem formulation and polynomial-time solution via KM algorithm
- Practical implementation that integrates well with existing MoE training systems
- Comprehensive empirical validation across different model scales and configurations
- Significant speedups achieved without compromising model performance

Main weaknesses:

- Limited discussion of scalability beyond 32 GPUs, though authors provided evidence that improvements do not diminish with scale
- Initial presentation of notations and problem formulation was dense, though improved with additional figures in revision
- Some questions about batch size limitations, though authors clarified typical training scenarios use small batch sizes per device

**Additional Comments On Reviewer Discussion:**

Outcomes from author-reviewer discussion:
The authors provided detailed responses addressing all major concerns:

- Clarified that speedups are robust to increasing GPU counts based on available results
- Added comprehensive analysis of data locality impact and communication volume statistics
- Explained practical constraints around batch sizes and gradient accumulation
- Provided additional experiments with varying GPU per node configurations
- Added discussion of typical industry deployment scenarios with 8 GPUs/node

Reviewer agreement:
All reviewers ultimately recommended acceptance after author responses. Initial scores ranged from 6-8, with final consensus at "accept" level

Suggestions to improve:

- Add more discussion of practical training scenarios and hardware configurations
- Include additional analysis of communication patterns and data locality
- Consider exploring more efficient alternatives to KM algorithm for cases requiring larger batch sizes

---

### Decision · Program_Chairs · 2025-01-22

Accept (Spotlight)